# RAPID*prep*: A Simple, Fast Protocol for RNA Metagenomic Sequencing of Clinical Samples

**DOI:** 10.3390/v15041006

**Published:** 2023-04-19

**Authors:** Rachel L. Tulloch, Karan Kim, Chisha Sikazwe, Alice Michie, Rebecca Burrell, Edward C. Holmes, Dominic E. Dwyer, Philip N. Britton, Jen Kok, John-Sebastian Eden

**Affiliations:** 1Centre for Virus Research, Westmead Institute for Medical Research, Westmead, NSW 2145, Australia; 2Sydney Institute for Infectious Diseases, School of Medical Sciences, The University of Sydney, Sydney, NSW 2006, Australia; 3PathWest Laboratory Medicine WA, Department of Microbiology, Nedlands, WA 6009, Australia; 4School of Biomedical Sciences, The University of Western Australia, Crawley, WA 6009, Australia; 5Departments of Infectious Diseases and Microbiology, The Children’s Hospital at Westmead, Westmead, NSW 2145, Australia; 6NSW Health Pathology Institute for Clinical Pathology and Medical Research, Westmead Hospital, Westmead, NSW 2145, Australia

**Keywords:** RNA sequencing, metagenomics, infectious diseases, diagnostics

## Abstract

Emerging infectious disease threats require rapid response tools to inform diagnostics, treatment, and outbreak control. RNA-based metagenomics offers this; however, most approaches are time-consuming and laborious. Here, we present a simple and fast protocol, the RAPID*prep* assay, with the aim of providing a cause-agnostic laboratory diagnosis of infection within 24 h of sample collection by sequencing ribosomal RNA-depleted total RNA. The method is based on the synthesis and amplification of double-stranded cDNA followed by short-read sequencing, with minimal handling and clean-up steps to improve processing time. The approach was optimized and applied to a range of clinical respiratory samples to demonstrate diagnostic and quantitative performance. Our results showed robust depletion of both human and microbial rRNA, and library amplification across different sample types, qualities, and extraction kits using a single workflow without input nucleic-acid quantification or quality assessment. Furthermore, we demonstrated the genomic yield of both known and undiagnosed pathogens with complete genomes recovered in most cases to inform molecular epidemiological investigations and vaccine design. The RAPID*prep* assay is a simple and effective tool, and representative of an important shift toward the integration of modern genomic techniques with infectious disease investigations.

## 1. Introduction

Despite major advancements in infectious disease diagnostics and treatment, infections remain a leading cause of death globally. Novel infectious agents and rapid pathogen evolution have led to considerable challenges for traditional diagnostics. At present, accepted methods for disease diagnostics rely on microbial isolation, targeted polymerase chain reaction (PCR), microarray-based assays, and serology [1]. As these traditional diagnostic methods are targeted, they are often limited in in their capacity to identify novel pathogens and co-infections. For example, although the reverse transcription polymerase chain reaction (RT-PCR) is both fast and relatively inexpensive, it often fails to detect novel organisms or where genetic variation occurs in the binding region of known pathogens targeted by primers or by probe [2]. Furthermore, many disease-causing agents are difficult to grow using culture-based methods or unculturable in vitro; accordingly, these approaches are inherently slow and limited for uncovering novel pathogen diversity. Indeed, such limitations in identifying and characterizing novel pathogens through routine pathology laboratories, as seen with severe acute respiratory coronavirus 2 (SARS-CoV-2) [3], remains one of the greatest global public health challenges. However, the impact is also significant at the level of individual care, where delays in diagnosis and treatment can dramatically affect clinical outcomes [4].

Advances in the cost and scale of genomic sequencing have provided important solutions to the challenges of emerging infectious diseases. Unbiased methods such as RNA-based metagenomic next-generation sequencing (RNA-mNGS) offer the capacity to recover and quantify sequences from pathogens with both DNA and RNA genomes [5], describe the microbiome and resistomes [6], and identify coinfections that may be associated with increased morbidity and mortality [7]. RNA-mNGS sequencing offers the unbiased detection of emerging pathogens with the greatest diagnostic potential, as it does not require any prior knowledge about the identity of the causative agent or its genomic sequence (i.e., it is cause-agnostic). The diagnostic capacity of RNA-mNGS was clearly demonstrated during the COVID-19 pandemic and the rapid identification of SARS-CoV-2 in less than a week after realization that the infections were likely caused by a novel agent [3]. The emergence of novel SARS-CoV-2 variants throughout the course of the pandemic and associated failures in RT-PCR primers for diagnostic [8] and whole genome sequencing [9] highlight the speed of pathogen evolution and the need for rapid and accurate unbiased sequencing. While RNA-mNGS is indeed powerful, there are some limitations when compared to traditional approaches. For example, the diagnostic sensitivity is lower compared to PCR or targeted enrichment due to the relatively low abundance of the viral sequences with respect to the high background from host or microbial nucleic acids [10]. Deeper sequencing may circumvent some of the limitations in sensitivity, although this approach is more costly and often time-consuming due to the turnaround times of higher-output sequencing platforms and thorough library QC requirements. Ultimately, this highlights the fact that the advancement of mNGS and targeted sequencing into clinical diagnostics will require the development of multiple tools to address multiple needs.

In response to emerging disease threats, there is a need for simple and fast RNA-mNGS approaches to provide rapid and reliable identification of pathogens in a timely manner to inform better treatment and control. Here, we developed and validated a streamlined RNA-mNGS method capable of detecting pathogen RNA (genomes and transcripts) starting from a sample collection, including sequencing and analysis in less than 24 h. This approach is largely based on day work for library prep and overnight sequencing on the Illumina iSeq. Furthermore, we developed the approach to utilize readily available reagents to ensure ease of access and reproducibility, particularly following the widespread adoption of amplicon-based whole-genome sequencing (WGS) during the COVID-19 pandemic. The RAPID*prep* assay is designed to be simple with minimal handling and QC requirements. However, it is still robust and includes all important steps, including genomic DNA (gDNA) removal and ribosomal RNA (rRNA) depletion, to boost sensitivity. We developed, optimized, and evaluated the utility of this approach on a range of clinical respiratory samples containing both known and unknown pathogens and compared the quantitative performance to quantitative RT-PCR. By providing real-time, high-resolution metagenomic data, the RAPID*prep* assay can inform the diagnosis of common and novel infections to control and monitor outbreaks.

## 2. Materials and Methods

### 2.1. Specimens

This study utilized common respiratory samples, including nasopharyngeal swabs and aspirates, along with cultured material of A/pdmH1N1 2009 influenza viruses and ZymoBIOMICS Microbial Community Standard (Zymo #D6300). The samples were specifically representative of a range of known viruses (SARS-CoV-2 and respiratory syncytial virus (RSV)), sample qualities (storage in standard viral transport medium (VTM) or Zymo DNA/RNA shield reagent), and extraction platforms (Roche MagNA Pure 96 Viral NA small volume, Zymo Quick-RNA Viral or ZymoBIOMICS DNA/RNA Miniprep Kits). The sample processing followed the manufacturers’ recommended protocols. The SARS-CoV-2 and RSV samples were quantified by RT-qPCR targeting the nucleocapsid [11] and nucleoproteins [12], respectively. Briefly, 5 µL of viral extract was converted to cDNA using the Invitrogen SuperScript IV VILO master mix before qPCR using IDT PrimeTime Gene Expression Master Mix with 500 nM and 250 nM of primers and probes, respectively. Finally, the study also included samples of unknown aetiology collected with parental consent from children with acute respiratory illnesses (mild and severe). This study was approved by the Sydney Children’s Hospitals Network (SCHN) human research ethics committee (HREC; approval numbers HREC/18/SCHN/263 and 2020/ETH00837) and the Western Sydney Local Health District HREC (approval numbers LNR/17/WMEAD/128 and SSA/17/WMEAD/129).

### 2.2. RAPIDprep Assay

The assay was divided into the following steps: gDNA removal; rRNA depletion; first strand cDNA synthesis; second strand cDNA synthesis and cleanup; tagmentation; library amplification and cleanup; and sequencing. A simple step-by-step protocol was made available from https://www.protocols.io/view/rapidprep-a-simple-fast-protocol-for-rna-metagenom-rm7vzbjkxvx1 (accessed on 22 December 2022). The specific reagents and their sources are listed in Table 1. For gDNA removal, 8 µL of sample extract (viral RNA, total DNA/RNA, or purified RNA) was combined with 1 µL each of Invitrogen 10X ezDNase Buffer and enzyme before 10 min incubation at 37 °C, then transferred to ice. For rRNA depletion, 1 µL of Qiagen FastSelect Mix (equally combined QIAseq FastSelect human, mouse, rat (HMR), bacterial 5S/16S/23S, and water) was added to the previous reaction before a step-wise incubation from 75 °C, 70 °C, 65 °C, 60 °C, 55 °C, 37 °C and 25 °C, held for 2 min at each step, then transferred to ice. For first strand cDNA synthesis, 4 µL of SuperScript IV VILO Master Mix (5X) and 5 µL of water were added to the previous reaction before incubation at 25 °C for 10 min, 50 °C for 20 min and 85 °C for 5 min, then transferred to ice. For second strand cDNA synthesis, 8 µL of Sequenase reaction buffer (5X), 1 µL diluted Sequenase enzyme (Sequenase dilution duffer and Sequenase v2.0 DNA Polymerase at a ratio of 2:1), and 11 µL of water are added to the previous reaction before incubation starting at 4 °C with a slow ramp (0.1 °C/s) to 37 °C for 10 min, then 95 °C for 2 min, then transferred to ice. The reaction was then topped up with a further 1 µL of diluted Sequenase enzyme before incubation at 37 °C for 30 min. The double stranded cDNA (ds-cDNA) was then purified using Omega Bio-tek Mag-Bind Total Pure NGS cleanup beads with a 0.8X bead to sample ratio and a final elution with 22 µL of Qiagen EB. The purified ds-cDNA (5 µL) was then prepared for sequencing using the Nextera XT DNA Library Preparation Kit with the IDT for Illumina–Nextera DNA unique dual indexing kit as per manufacturer’s instructions except for the following modifications: 16X cycles was used for library amplification followed by purification with Omega Bio-tek Mag-Bind Total Pure NGS cleanup beads using a 0.8X bead to sample ratio, and a final elution with 32 µL of Qiagen EB. Library QC was then performed using a High Sensitivity D1000 ScreenTape on the Agilent 2200 TapeStation system with gating of the fragments between 200 bp and 700 bp, before final dilution to 0.1 nM for loading and sequencing on an Illumina iSeq (paired-end 150 bp sequencing). As the minimal sequencing yield for each library should be 1 million paired reads, 1–4 libraries can be multiplexed per iSeq run. For our large, batched run, we prepared and indexed 39 samples and one no-template control (NTC). These were pooled evenly and sequenced on an Illumina NovaSeq SP 300 cycle lane generating at least 4 million paired reads per library (NCBI SRA SRR22726217 SRR22726256).

### 2.3. Development of Final Assay Conditions

We explored three aspects of optimizing the RAPID*prep* assay that were focused on simplifying the protocol to improve turnaround time and determining the optimal yield of the final libraries. These included testing: (1) rRNA depletion performance; (2) ds-cDNA yield; and (3) number of cycles for library amplification. For the rRNA experiments, the standard pre-cDNA hybridization step (as above) was compared with a simplified approach, spiking 1 µL of depletion oligos (FastSelect mix) directly into the first strand cDNA reaction with the relative amount of rRNA following sequencing measured as output. For the ds-cDNA yield experiments, the standard Sequenase two-step reaction was compared with a single-step reaction combining the total amount of Sequenase enzyme (2 µL) and reaction time (40 min extension at 37 °C). The output was measured by Agilent TapeStation to compare the library yield of each approach. For the library amplification experiments, we titrated the number of indexing PCR cycles between 14X to 20X in two cycle steps. The output was also measured by Agilent TapeStation to compare the library yield of the different cycles; however, the libraries were also sequenced to determine the sequence read duplication rate. For all the experiments, the same three respiratory sample extracts (clinical nasopharyngeal swabs collected in Zymo DNA/RNA shield and extracted with both the Zymo Viral RNA and ZymoBIOMICS DNA/RNA miniprep kits used along with an NTC). Samples were run in duplicate with the mean and standard deviation values reported.

### 2.4. Severe Acute Respiratory Infections in Children Cohort

A subset of the samples the severe acute respiratory infections (SARI) in hospitalized children had been previously sequenced using a commercial RNA sequencing assay (NCBI SRA SRR22838411 SRR22838442). These data were used to compare against libraries made using the RAPID*prep* assay (Appendix A). Briefly, these RNA samples were prepared for sequencing using the SMARTer Stranded Total RNA-Seq Kit v2 Pico Input Mammalian with unique dual indexes (Takara Bio, Kusatsu, Japan) as per the manufacturer’s instructions, and sequenced on an Illumina NovaSeq with at least 40 million paired reads for the library.

### 2.5. Bioinformatic Analysis of RNA-mNGS Data

Raw sequence reads were first quality trimmed and filtered using FastP v0.19.6 [13] with default parameters, except the read length filter was 50 bp. Read duplication rates (i.e., quantifying identical sequences) were extracted from FastP quality reports. The trimmed reads were then mapped to the human genome using STAR-aligner v2.6.1b [14], followed by Burrows–Wheeler Aligner (BWA) v0.7.17 [15] to ensure complete human sequence removal. The trimmed, human and non-human reads were then filtered into rRNA and non-rRNA and quantified using SortMeRNA v2.1b [16] with the default clustered 5S, 5.8S, 16S, 18S, 23S, and 28S databases (available from: https://github.com/sortmerna/sortmerna/tree/master/data/rRNA_databases, accessed on 12 April 2023); then, the trimmed, non-human, non-rRNA reads were de-novo-assembled using Megahit v1.1.3 [17] before annotation using blast+ v2.11 [18] and diamond v2.0.11 [19] with default e-value thresholds against the NCBI GenBank database (retrieved on 4 August 2021). A read-based analysis was also performed of the trimmed, non-human, non-rRNA datasets by mapping against the microbial taxonomic database in MetaPhlAn v3.0.13 [20]. Comparative analysis of microbial abundance was performed using calculated z-scores in R v3.4.3. Final viral read counts were also determined by alignment of trimmed, non-human, non-rRNA reads to the de-novo-assembled contigs and/or known viral reference genomes for the SARS-CoV-2, influenza virus and RSV samples using BBMap v 37.98 [21]. Maximum likelihood trees for individual viruses were estimated using PhyML v2.2.4 [22] with the GTR + Gamma substitution model and 1000 bootstrap replicates.

## 3. Results and Discussion

The aim of this study was to develop a simple yet robust workflow for RNA-mNGS of clinical samples that can provide a cause agnostic laboratory diagnosis in less than 24 h. The RAPID*prep* assay is comparable to other meta-transcriptomic assays, in that it aims to unbiasedly sequence the non-host, non-rRNA RNA for pathogen detection and quantification. However, it is unique in its simplicity, with reduced handling and a uniform protocol for sequencing across a range of sample types, qualities, and quantities. The first steps aimed to remove gDNA and rRNA to improve target sensitivity before random double-stranded cDNA synthesis and amplification. Minimizing the processing and handling was important to ensure that the entire protocol could easily be completed in less than 6 h. This was primarily achieved by the basic assay design, with most steps being additive and performed in a single tube without the need for reaction clean-ups (bead-based purifications). However, we explored this further by attempting to simplify the rRNA-depletion and ds-cDNA synthesis steps and optimizing the library amplification yield through a range of experiments using three representative respiratory samples (RESP01-RESP03).

### 3.1. Development of the RAPIDprep Assay

#### 3.1.1. rRNA Depletion

rRNA is the most abundant component of total RNA isolated from eukaryotic and microbial cells [23]. While the importance of rRNA-depleted libraries for improved coverage of mRNA for transcriptome sequencing is recognized [24,25,26], it is particularly important for other non-rRNA targets, such as viruses with RNA genomes. This enrichment of non-rRNA by rRNA depletion enables better identification and genome recovery of viral pathogens with RNA genomes, such as coronaviruses, influenza viruses, and paramyxoviruses that are emerging disease threats. The FastSelect reagent blocks transcription with proprietary probes that bind to mammalian and microbial rRNA. As such, it does not necessarily deplete rRNA, but rather prevents its synthesis during cDNA steps. To increase the speed of the protocol, we sought to determine if the FastSelect probes could be added directly to the first strand cDNA synthesis step without the need for pre-cDNA hybridization step that added approximately 30 min of reaction and handling time. The relative abundance of rRNA in the final sequenced library between the two approaches was compared (Figure 1A).

A clear trend was observed across the three samples, where the final libraries made using a dedicated pre-cDNA hybridization step had a dramatically smaller proportion of rRNA in the final library yield. To further investigate the effect of the rRNA depletion method on the final library composition, we examined the residual rRNA by kingdom, and their relative abundance was once more compared across different samples and methods (Table 2). Similarly, all classes of rRNA were better depleted utilizing the pre-cDNA synthesis hybridization protocol with the residual rRNA not exceeding 4.0% (RESP03 bacterial 23S rRNA), and in most cases less than 1.0%, while the in-reaction approach had up to 22.0% residual rRNA (RESP02 eukaryotic 18S rRNA). FastSelect is a simple and effective solution for the removal of rRNA, although the probes clearly require dedicated steps to hybridize efficiently and, in this case, must occur prior to first strand cDNA synthesis. While performing the rRNA step prior to cDNA hybridization increases the total protocol time slightly, the greatly improved rRNA depletion outweighs this and improves the sensitivity of the overall assay for better pathogen detection.

#### 3.1.2. Double-Stranded cDNA Synthesis

As the purpose of this method was to be robust yet as rapid as possible, we next explored the feasibility of reducing the second-strand synthesis of cDNA from a two-step process to one-step. Sequenase enzyme is a modified bacteriophage T7 DNA polymerase that lacks 3′→5′ exonuclease activity with improved processivity and speed [27]. The standard reaction occurs in two steps, where initial double-stranded cDNA from the first-strand reaction will be produced from randomly primed single-stranded DNA template [28]. This will be followed by the addition of a further enzyme for final extension of the ds-cDNA products. We sought to compare this two-step approach to a simplified one-step protocol, where the extension time and enzyme concentration during the first part was increased to match the overall two-step approach. Not only would this shorten the workflow by up to 15 min—it would also help minimisze the handling and potential opportunities for contamination. However, across the three test samples, we saw lower total library yields using the simplified one-step method (Figure 1B). While the yields of the one-step protocol were sufficient for sequencing, the desire for a single uniform protocol across varying sample types and qualities favored, here, the approach with the greatest yield. Therefore, like the rRNA depletion optimization and despite a small trade-off in time and handling, our final assay utilized the two-step protocol that provided greatest performance.

#### 3.1.3. Library Amplification

It is widely accepted that library preparation can introduce systematic bias to the characterization and representation of microbial communities in a sample [29,30,31,32]. Bias is most readily introduced during the library amplification stage. Some studies argue that the simplest means to mitigate this bias during PCR is to avoid library amplification altogether. For the RAPID*prep* assay, a PCR-free library protocol would likely be unattainable due to the low concentrations of input total nucleic acid, particularly from swabs and cell-free viral samples. Illumina Nextera XT is a commercially available library preparation kit that uses a transposase-based *tagmenention* reaction to fragment and add adapters onto template dsDNA [31]. Following this, limited-cycle PCR is used to barcode and complete index adapters before sequencing [33]. While other library preparation kits would be compatible here, Nextera XT is simple and fast and, therefore, an ideal partner for the RAPID*prep* assay. Furthermore, Nextera XT is widely used for amplicon-based WGS of viral pathogens [34,35,36] and offers potentially greater adoption compared to other library preparation kits. As low input total nucleic acid necessitates PCR amplification, we sought to identify an optimal cycle number that afforded greatest library yields while limiting potential amplification bias. A titration experiment was, therefore, performed, with final library yield measured using DNA molarity as determined by Agilent Tapestation (Figure 1C), and sample bias was measured by calculating the read duplication rate of the sequenced libraries (Figure 1D). Percentage duplication rate is an ideal proxy for sequence bias, as the redundant reads are typically introduced during library amplification PCR. Furthermore, duplicate reads limit the entropy of the final dataset and potentially introduce bias for both sample identification and, particularly, for quantification [37].

Here, we explored the optimal cycle conditions from 14X to 20X PCR cycles (Figure 1C,D). For two samples (RESP01 & RESP03), the DNA yield was greatest at 16X cycles, while for one sample (RESP02) it was 18X cycles. (Figure 1C) The apparent trend showing reduced yield at cycles 18X and above was due to overamplification-induced artefacts with fragments exceeding the upper range (1000 bp) of the Tapestation analyzer (data not shown). While such large fragments are likely to be sequenced when denatured, they present a challenge for quantifying and final library QC. Similarly, the percentage duplication data demonstrated a clear increase in overamplification of libraries at cycles 18X and above for all three samples (Figure 1D). A high PCR duplication rate cannot simply be overcome using deeper sequencing methods; this is a fundamental issue that can only be mitigated at the time of library preparation. Indeed, read duplicates can only be identified post-sequencing; for this reason it is advantageous to choose PCR cycles that maximize yield while minimizing duplication rates. Here, this optimum seemed to be 16X PCR cycles, which were used for the final RAPID*prep* assay. However, we note that these conditions are favorable for commonly used respiratory samples such as nasopharyngeal swabs, and further optimization would be required for samples of different biomass and pathogen loads.

### 3.2. Application of the RAPIDprep Assay to a Panel of Respiratory Samples

To provide a broad assessment of RAPID*prep* performance, we selected a range of respiratory samples and control material (n = 40) for a combined, proof-of-concept run, using the optimized protocol (Table 3). These samples varied in microbial composition, sample collection, quality, and extraction, and were designed to reflect a broad snapshot of real-world sampling performance. Libraries *RAPID01-12* were derived from SARS-CoV-2 positive respiratory swabs collected and processed within one week following collection from a household transmission study. *RAPID13* and *RAPID14* were viral stocks collected from A/pdmH1N1 2009 influenza virus infected cells. Further known positive samples were prepared as libraries *RAPID25-32* that were RSV-positive and extracted through a diagnostic pathology service using a high-throughput bead-based platform (Roche MagNA Pure). *RAPID15* and *RAPID16* were high-quality cultured materials containing standard amounts of known bacteria and fungi (ZymoBIOMICS Microbial Community Standards) and were process controls for a study investigating unknown SARI in hospitalized children. A subset of these SARI samples (libraries *RAPID17-24*) was included here, as they represented residual, and often highly degraded, specimens collected through routine diagnostic services and had existing deep-sequencing data for comparison. Finally, high-quality respiratory samples of unknown etiology collected in Zymo DNA/RNA shield were used (*RAPID33-39*) along with an NTC reaction (*RAPID40*). As per the protocol, no specific sample QC was performed and 8 µL of neat extract (total NA, viral RNA, or RNA) was used as input based on the protocol set out in Section 2.2. All 40 samples produced libraries with a mean yield of 31.0 nM (range: 1.0 nM to 134.5 nM) that were pooled equally and sequenced on a single Illumina NovaSeq SP lane (Table 3). Of note, we found a poor correlation between intermediate ds-cDNA concentration (ng/uL) and library yield (nM), R^2^ = 0.19 (Appendix A), suggesting that input into the library amplification steps was largely concentration-independent. The mean sequence yield per library was 15,577,978 reads (range: 9,453,054 to 28,187,178 reads).

Following sequencing, each library was then analyzed for low-quality, human, and rRNA content before taxonomic assignment and quantification using a standard mNGS pipeline. The sequence reads for each library were filtered into five specific categories: low quality, human rRNA, human non-rRNA, non-human rRNA, and non-human non-rRNA, and the relative proportions were compared across the sample set and groups (Figure 2). Low-quality reads were of the highest abundance in the samples from the SARI cohort (*RAPID17-24*), where the mean low-quality reads were 23.4% of total libraries. These were “rescued” diagnostic specimens that had gone through multiple freeze–thaws in the pathology laboratory, with many likely heavily degraded. Suboptimal sample quality also likely explains some of the variation in low-quality reads in the SARS-CoV-2 cohort (*RAPID01-12*), with delayed transport to the laboratory following collection at home. Low-quality input samples generally result in an increased amount of homopolymers and short-fragment reads, and are hallmarks of endpoint sample degradation that can be used as a measure of sample quality [38]. While sample quality is an issue, low-biomass samples would be expected to have higher levels of low-complexity reads that will be removed during the initial QC steps, such as the NTC library (*RAPID40*).

The proportion of reads that mapped to the human genome had a mean value of 70.8% across all the libraries and sample types; however, this varied widely (range: 20.0% to 98.4%). Sequencing data from host-associated microbes may contain host cells, usually acquired at the time of sampling [24]. Several factors can influence the abundance of host material at the point of collection, including the collection route, the sampling device (such as flocked vs. non-flocked swabs), the technique, and the collector experience [26]. Furthermore, as the human genome is significantly larger than microbial and viral genomes, host-derived nucleic acids can easily be over-represented, even if in relatively small amounts. On average, human reads were the most common read assignments across all samples, except for the SARI cohort (*RAPID17-24*) and NTC (*RAPID40*). The collection method of samples in the SARI cohort varied, and many of these samples were acquired from sources other than nasopharyngeal swabs, including aspirates, due to age and hospitalization, which likely contributed to the variable yields. Contamination of sequencing data by human nucleic acid can readily occur, with putative sources including adjacent samples or from the collector [39]. The relative abundance of human reads can also be affected by sample processing steps, including the extraction kit used. For example, the highest levels of human RNA were found in the RSV-positive respiratory swabs (Figure 2) that were processed using the Roche MagNA Pure 96 Viral NA small volume kit. In contrast to the other Zymo extraction kits, this platform includes an initial Proteinase K digestion that likely increases the overall extraction and therefore relative yield of human DNA and RNA [40]. Clearly, sample collection, sample quality, and the extraction approach all contributed here, and generally contributed to the variability in the identification and sensitivity of pathogens by mNGS. Our use of real-world samples limits the conclusions about which factors contribute more; however, it is important these factors be considered when implementing mNGS, such as the RAPID*prep* assay locally. High levels of human sequences not only limit the sensitivity of target non-human, non-rRNA, but also increase the risk of residual human DNA being deposited in public archives, which presents an ethical concern and potentially indefinable information. Care must be taken at the point of sampling or in processing to reduce the amount of unnecessary human tissue acquired. Finally, we also note that care must be taken during the informatic steps to ensure host read filtering, not only for patient privacy but also to ensure that pathogen sequences are not inadvertently filtered due to close homology with the host genome, as this would impact assay sensitivity. This should be optimized by mapping to the host genome with appropriate stringency.

Overall, residual rRNA was limited across the sequence libraries, highlighting the performance of the RAPID*prep* assay where the mean non-human rRNA was only 1.3% (Figure 2). However, despite our extensive optimization experiments (Figure 1A), rRNA depletion remained incomplete in some samples, such as *RAPID02*, *RAPID15*, and *RAPID16* that contained between 9.7% and 9.9% non-human rRNA reads of each library. One reason could be the limited microbial diversity captured by rRNA depletion probes, such as Qiagen FastSelect and equivalent products. In both *RAPID15* and *RAPID16*, taxonomic assignment of the residual rRNA showed a predominance of *Bacillus spp* (63.0% to 73.0% rRNA reads), while the same organisms were at much reduced abundance in non-rRNA data (18.0% to 22.0% non-rRNA reads). Such an imbalance was not noted for other taxa present in the mock community, suggesting some failure in the targeting of the *Bacillus* spp. by the FastSelect probes. However, in these same libraries, residual human rRNA was also present, suggesting more likely that the level of FastSelect probes and, for higher input RNA such as these cultures or even whole tissue, the concentration might need to be increased (Figure 2). While the overall rRNA-depletion performance was good, incomplete rRNA depletion limits the detection of target species [41]. The overarching goal of this and other meta-transcriptomic approaches is to produce sufficient non-human non-rRNA reads to identify pathogens of interest. Where non-human, non-rRNA reads are unexpectedly low, the depth of sequencing becomes an important consideration. Across the samples in this study, the mean number of non-human non-rRNA reads was 3,125,035, which would be considered an acceptable read number for pathogen identification [41]. However, the lowest yielding non-human non-rRNA library was *RAPID30*, with only 5898 reads. At this sequencing depth, potentially no pathogen sequences will be identified, and it is also difficult to rule out infections (often a goal of clinical mNGS); therefore, target yields of >1M non-human, non-rRNA would be ideal.

### 3.3. Viral Sequence Identification, Genome Recovery, and Quantitative Performace

For each library, the non-human, non-rRNA sequences were taxonomically assigned and quantified with a focus on viral reads expressed as log-transformed reads per million (_log_RPM) values (Table 4). In samples where known pathogens were detected, e.g., SARS-CoV-2, RSV, and the A/pdmH1N1 influenza virus, the _log_RPM values ranged from 2.64 to 6.00. The mean _log_RPM values for the sample groups were 5.17, 3.74, and 5.88, respectively, and in all samples the expected respiratory pathogen was identified. In addition to detecting known pathogens, we sought to evaluate the utility of the RAPID*prep* assay in identifying unknown pathogens in two sample groups. The first group was a SARI cohort (*RAPID17-24*) and the second group included children with mild respiratory infections (*RAPID33-39*) (Table 3). For the SARI cohort, two samples returned a positive result using the RAPID*prep* assay, and provided for the identification of possible causative pathogens that had not been identified using conventional diagnostic methods (Table 4). Human cytomegalovirus (CMV) was identified at low levels in *RAPID17*, with a _log_RPM value of 0.62, which mapped to multiple viral genes and were likely true hits and not host-derived. Interestingly, the abundant Influenza C virus was identified in *RAPID22* with a _log_RPM value of 5.77 (Table 4). As this sample group was comprised of samples stored for up to six weeks at 4 °C before transfer to −80 °C, and also thawed and refrozen multiple times, the subsequent identification of possible pathogens emphasizes the clear diagnostic potential of the RAPID*prep* assay and RNA-mNGS approaches. Across the mild unknown cases, human rhinovirus was detected in five of seven samples, with _log_RPM values ranging from 4.79 to 6.00 (Table 4). Incidentally, human betaherpesvirus 7 (HHV-7) was also detected in *RAPID36*; although it was not likely responsible for the acute respiratory illness symptoms, it remains an important detection.

To assess the genome recovery of the RAPID*prep* assay, we examined the sequence coverage of the SARS-CoV-2 (*RAPID01-12*) and RSV libraries (*RAPID25-32*) by mapping against the viral genome. For the SARS-CoV-2 data, all libraries (n = 12) produced genome coverage > 99.9% at a mean depth of 7270X (range: 22X to 23,085X). For the RSV data, only half the libraries (n = 4) produced genome coverage > 90%, while the remaining libraries ranged from 41.9% to 77.8%. The reduced genomic recovery was due to lower coverage depth (mean: 7X, range: 1X to 23X) (Appendix A). As mentioned, the reduced genomic yield in the RSV samples was due to an over-abundance of human sequences (Figure 2). Despite this, the genomic recovery was more than sufficient to subtype both the SARS-CoV-2 and RSV cases, as well as the previously undiagnosed rhinovirus sequences from the unknown mild infections, using a phylogenetic approach (Appendix A). This not only demonstrates the diagnostic performance of the RAPID*prep* assay, but also the utility to allow further epidemiological investigation of potential pathogens. Indeed, the identification of two distinct rhinovirus C strains in RAPID39 (Appendix A) highlights the assay’s capacity to sequence and identify not only mixed infections by different viruses, but also closely related strain or variants. This genomic data could also be used to design new diagnostic assays and inform vaccine development, as seen during the COVID-19 pandemic with the sequencing and release of the first SARS-CoV-2 genome [3].

To assess the quantitative performance of the RAPID*prep* assay, we utilized the cycle threshold values generated using RT-qPCR and compared these to the _log_RPM values from the RAPID*prep* SARS-CoV-2 and RSV positive libraries (Figure 3). The SARS-CoV-2 sample group comprised 12 PCR positive samples (*RAPID01-12*). Here, we identified SARS-CoV-2 in all 12 samples using the RAPID*prep* method and identified a strong linear relationship (R^2^ = 0.86) between _log_RPM and CT values (Figure 3A). As expected, _log_RPM increases as CT values decrease, indicating that the RAPID*prep* method was sensitive to relative viral load. A similar result was observed for the RSV data (*RAPID25-32*) where the read _log_RPM and RT-qPCR CT values were well-correlated (R^2^ = 0.85) (Figure 3B). Together, these results highlight the quantitative performance of the RAPID*prep* assay and indicate that normalized relative abundance, such as RPM values, can be used as estimates of the initial viral load of known and unknown pathogens.

### 3.4. Comparison of RAPIDprep to Commercial Assay

Finally, we compared the performance of the RAPID*prep* assay against a commercial assay Takara SMARTer Stranded Total RNA-Seq Kit v2. The SMARTer-Seq libraries were prepared previously from eight residual diagnostic samples and two Zymo mock-community controls as part of a study into the possible infectious causes of SARI in children (Table 2). These libraries, labeled as *ICU15-24*, had the same source RNA extracts for the RAPID*prep* libraries (*RAPID15-24*), with each RNA labeled with the same sample number (i.e., ICU15 and RAPID15 share the same RNA source; see Appendix A). For the analysis, we processed each library by the removal of low-quality, human, and non-human rRNA sequences before extracting 1M non-human rRNA for alignment and taxonomic assignment using MetaPhlAn3. An unclustered heatmap of microbial abundance (Z-score for the top 24 taxa) was used to compare the sensitivity and specificity of the mNGS protocols (Figure 4). Overall, there was good concordance between the SMARTer-Seq and RAPID*prep* assays, with conservation of nasal–oral taxa across protocols, particularly for the predominant species (*ICU/RAPID17*). Furthermore, the Zymo mock-community control samples displayed good repeatability across methods, presenting similar row z-scores. As anticipated, there was some variation between methods, which was likely due to the depth of sequencing and batch effects. For example, the increased abundance of *Escherichia coli* sequences across the RAPID*prep* libraries indicates a common source, most likely from using different reagents. This highlights the need for positive and non-template controls, as well as reagent batching when performing mNGS studies, particularly with low-biomass samples [42]. Finally, the influenza C virus detected in *RAPID22* was also identified in *ICU22* (Table 4), again confirming that the diagnostic value was largely comparable. The SMARTer-Seq protocol is slower (2 day protocol) and more costly (~2X), but it is designed for very low inputs (RNA amounts < 1 ng) and, therefore, it is more suitable for mNGS of low-biomass sample types such as cerebrospinal fluid (CSF), where it has been used for pathogen discovery [43]. This aspect of the RAPID*prep* assay is yet to be explored.

### 3.5. Study Limitations

Here, we presented a simple yet robust workflow for the rapid mNGS of RNA from clinical respiratory samples. Despite demonstrating the utility of the RAPID*prep* assay across a range of sample types and pathogens, several limitations of this study remain. First, the major development of the assay was restricted to a limited set of respiratory samples; therefore, the performance across the broad range of sample types typically encountered in a pathology setting has yet to be rigorously explored. However, the RAPID*prep* has been used successfully in our hands using RNA from blood, solid tissue, stool, and cerebrospinal fluid (CSF). The only modifications made included limiting the library amplification cycles for high biomass samples tissue, blood, and stool to 14X or extending the cycles for low biomass samples of CSF up to 18X. Together with Appendix A, showing that ds-DNA concentrations did not predict final library yield, these results show that initial sample biomass and quality are the most likely factors shaping successful library preparation; however, this can be largely overcome by grouping samples in biotypes and expected biomass, rather than performing detailed sample QC. This is consistent with the original aim of the assay, which was to be a rapid response tool (for emerging disease threats or even priority patient care), rather than a replacement for rigorous accredited testing, even using other mNGS assays.

A second limitation is that our RNA sequencing approach, using an initial gDNA removal step, limits the detection of DNA viruses (for example adenovirus, polyomavirus, and parvoviruses) to those that are transcriptionally active; i.e., free virus particles containing gDNA will not be sequenced. This can be overcome by the parallel sequencing of gDNA using the sample protocol above but skipping the gDNA, rRNA depletion, and ds-cDNA synthesis steps. However, given the focus on identifying disease-causing agents, simply and rapidly, we focused here on a “single-test approach” and detecting RNA of infecting (and likely replicating) organisms, which should provide the greatest coverage across pathogens with both RNA and DNA genomes.

Finally, it is remains unclear how RNA-based mNGS approaches such as RAPID*prep* might replace routine methods such as RT-PCR. Here, we showed quantitative performance where read levels were proportional to RT-PCR CT values (Figure 3); however, we did not determine the ultimate sensitivity and the limits of detections. Similarly, the depth of sequencing required to ensure a negative result remains to be determined for mNGS, and while we proposed 1M non-human, non-rRNA sequences to capture the broad microbial and pathogen diversity, this threshold is yet to be empirically determined and is an important avenue for further research.

## 4. Conclusions

The RAPID*prep* assay was designed specifically as a rapid response tool, and it has proven to be effective in novel disease investigations, including identifying the first cases of the emergent Japanese encephalitis virus during the 2021-22 outbreak in south-eastern Australia [44,45,46] and characterizing the first cases of COVID-19 in NSW [35]. The assay has also been used to investigate non-human diseases, and it was critical to the genome recovery of a novel Hendra virus variant detected initially from a fatal equine infection by pan-paramyxovirus RT-PCR [2]. In the future, pathogen-agnostic mNGS testing will likely assume a greater role in identifying and quantifying novel, emerging, and re-emerging pathogens to guide individual patient management and public health responses as part of communicable disease control.

## Figures and Tables

**Figure 1 viruses-15-01006-f001:**
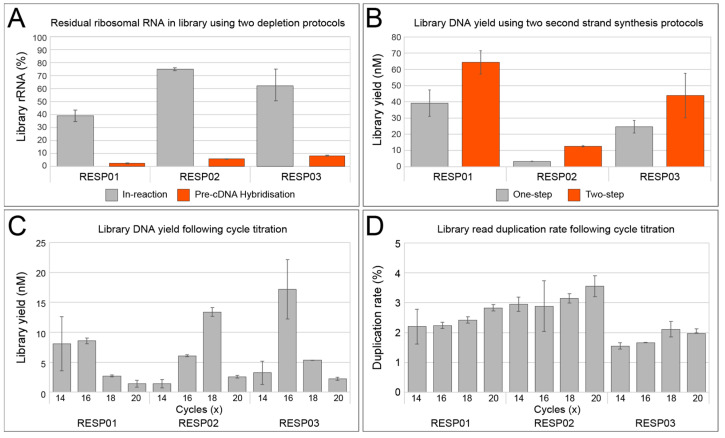
RAPID*prep* development experiments. All results here are derived from the same sample extracts (RESP01-RESP03) run in duplicate and presented as mean values and error as standard deviation (SD). (**A**) The shaded bars are representative of the percentage of residual rRNA reads in the library following rRNA depletion with either an in-reaction cDNA synthesis method (grey) or a pre-cDNA hybridization approach (orange). The bars are clustered with respect to the sample they are derived from, labeled on the X-axis. (**B**) A comparison in total library yield, in nanomolar generated using Tapestation values, following a parallel experiment with a one-step and two-step second strand synthesis step using the Sequenase enzyme. The grey and orange shaded bars are representative of the one-step and two-step protocols, respectively. (**C**) Grey-shaded bars represent the total library yield of each sample under different library amplification cycling conditions. The X-axis is marked with the number of amplification cycles and is sub-grouped by source sample. (**D**) The duplication rate of reads generated in the final libraries following cycle titration; the number of cycles for each sample is indicated on the X-axis, and is sub-grouped by source sample.

**Figure 2 viruses-15-01006-f002:**
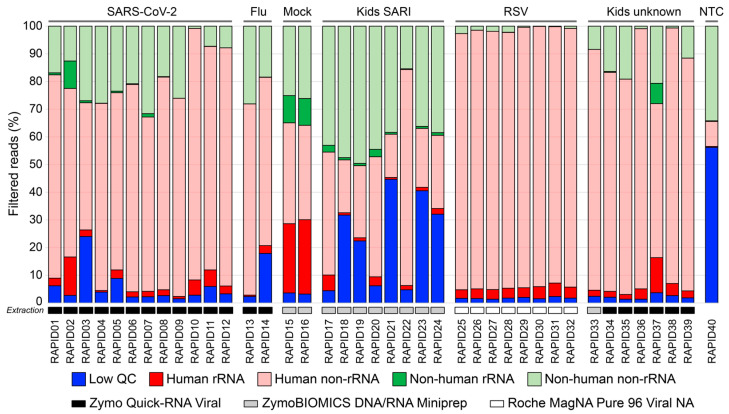
Filtered read distribution and classification across forty RAPID*prep* libraries. The sequence reads were classified into five categories: low-quality reads (blue), human rRNA reads (red), human non-rRNA (pink), non-human rRNA reads (green), and non-human non-rRNA reads (light green). Low-quality, human rRNA, human non-rRNA, and non-human rRNA were excluded from downstream analysis, and the non-human non-rRNA reads were the sole target reads for pathogen detection. Relative distribution was calculated by dividing the number of reads mapping to the relative category by the total number of reads for the individual library, before conversion into a percentage by multiplying the value by 100. The results were ordered by library number and grouped by sample type with a further key in grey shaded indicating the sample extraction platform used.

**Figure 3 viruses-15-01006-f003:**
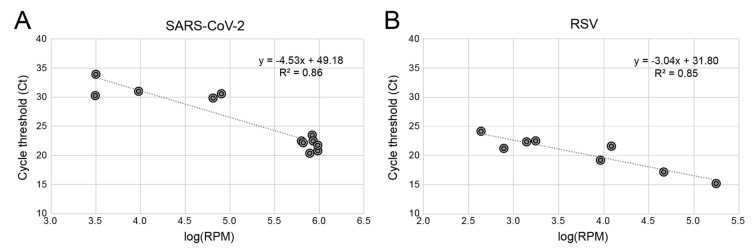
Quantitative detection of SARS-COV-2 and RSV sequences. A simple linear-regression model was applied to both SARS-CoV-2 (**A**) and RSV (**B**) data sets with a line of best fit estimating the relationship between log-transformed reads per million (_log_RPM) and cycle threshold (CT) values. The linear-regression slope coefficient and the intercept parameter are printed on the top right of each plot, with R^2^ calculated to measure the goodness of fit.

**Figure 4 viruses-15-01006-f004:**
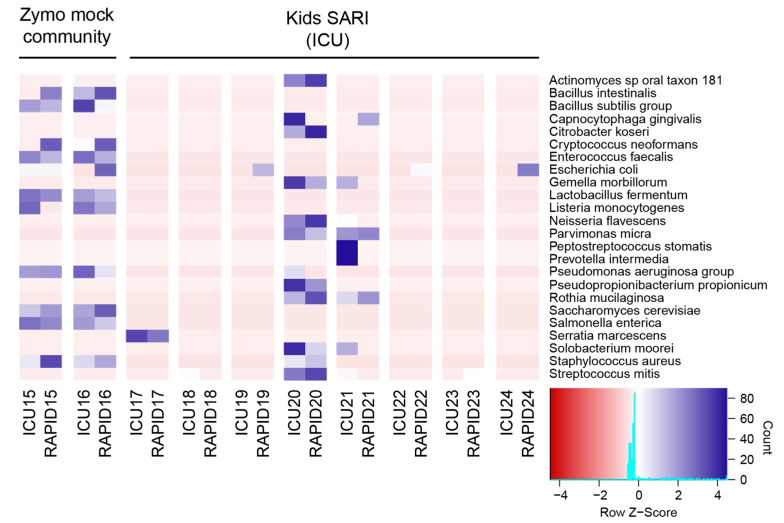
Comparison of RAPID*prep* to commercial RNA library preparation kit. Using previously generated data for the kids SARI cohort, we compared the 24 most abundant species identified across both protocols for the same set of samples. An unclustered heatmap of microbial abundance (Z-score) is shown, with differences between samples identified by a deeper blue shading, while organisms conserved across samples are lighter blue through to red. A frequency histogram is overlayed on the color key and signifies the count of each Z score at any given point. Tick labels on the X-axis in the ICUXX format represent deep-RNA sequencing generated previously, while tick labels in the RAPIDXX format represent sequencing data generated in this study using the RAPID*prep* assay for the corresponding samples.

**Table 1 viruses-15-01006-t001:** Reagents used for RAPID*prep* assay.

Reagent	Catalogue	Supplier
QIAseq FastSelect-rRNA HMR	334385	Qiagen, Hilden, Germany
QIAseq FastSelect–5S/16S/23S	335921	
Invitrogen SuperScript IV VILO Master Mix	11756050	Thermo Fisher, Waltham, MA, USA
Sequenase Version 2.0 DNA Polymerase	70775Y200UN	
Invitrogen ezDNase Enzyme	11766051	
Mag-Bind^®^ TotalPure NGS	M1378-01	Omega Biotek, Norcross, GA, USA
Nextera XT DNA Library Preparation Kit	FC-131-1096	Illumina, San Diego, CA, USA
IDT^®^ for Illumina DNA/RNA UD Indexes	20027213	
iSeq 100 i1 Reagent v2 (300-cycle)	20031371	

**Table 2 viruses-15-01006-t002:** Relative abundance of archaeal, bacterial, and eukaryotic rRNA using two different approaches, with red shading indicating greater read depth as per the key provided at the bottom of the table.

	In-Reaction						Pre-cDNA Hybridisation					
rRNA	RESP01		RESP02		RESP03		RESP01		RESP02		RESP03	
Archaeal:16S	3.5%	2.9%	7.5%	7.3%	6.1%	4.0%	0.0%	0.0%	0.1%	0.1%	0.2%	0.1%
Archaeal:23S	10.9%	9.5%	19.2%	19.9%	22.0%	16.2%	0.1%	0.1%	0.8%	0.7%	1.6%	1.4%
Bacterial:5S	0.7%	0.8%	0.2%	0.2%	0.4%	0.6%	1.0%	1.3%	0.9%	0.8%	1.4%	1.3%
Bacterial:16S	0.7%	0.6%	2.1%	2.1%	3.0%	2.2%	0.1%	0.0%	0.2%	0.2%	0.2%	0.1%
Bacterial:23S	3.5%	3.1%	10.1%	10.7%	20.7%	18.1%	0.2%	0.3%	2.2%	2.1%	4.0%	3.9%
Eukaryotic:5.8S	0.5%	0.5%	1.1%	1.1%	0.9%	0.8%	0.0%	0.0%	0.1%	0.1%	0.1%	0.1%
Eukaryotic:18S	14.0%	11.9%	22.0%	22.1%	11.2%	7.7%	0.3%	0.4%	0.9%	0.8%	0.6%	0.5%
Eukaryotic:28S	8.5%	6.7%	12.6%	13.0%	6.6%	4.4%	0.2%	0.2%	0.7%	0.7%	0.4%	0.4%
**rRNA levels**	0.0%	5.0%	10.0%	15.0%	20.0%	25.0%						

**Table 3 viruses-15-01006-t003:** RAPID*prep* sample summary overview table.

Library.	Group	Virus	Type	Extraction Method	Library Yield (nM)	Data Output (Reads)
RAPID01	COVID-19	SARS-CoV-2	Nasopharyngeal swab	Zymo Quick-RNA Viral	8	16,810,302
RAPID02		SARS-CoV-2	Nasopharyngeal swab	Zymo Quick-RNA Viral	34.7	11,620,222
RAPID03		SARS-CoV-2	Nasopharyngeal swab	Zymo Quick-RNA Viral	2.8	18,322,864
RAPID04		SARS-CoV-2	Nasopharyngeal swab	Zymo Quick-RNA Viral	2.4	12,707,642
RAPID05		SARS-CoV-2	Nasopharyngeal swab	Zymo Quick-RNA Viral	2.2	15,327,662
RAPID06		SARS-CoV-2	Nasopharyngeal swab	Zymo Quick-RNA Viral	13.2	15,271,010
RAPID07		SARS-CoV-2	Nasopharyngeal swab	Zymo Quick-RNA Viral	3.1	11,147,058
RAPID08		SARS-CoV-2	Nasopharyngeal swab	Zymo Quick-RNA Viral	6.1	9,453,054
RAPID09		SARS-CoV-2	Nasopharyngeal swab	Zymo Quick-RNA Viral	15.2	17,326,098
RAPID10		SARS-CoV-2	Nasopharyngeal swab	Zymo Quick-RNA Viral	9.1	15,531,486
RAPID11		SARS-CoV-2	Nasopharyngeal swab	Zymo Quick-RNA Viral	1.6	10,903,012
RAPID12		SARS-CoV-2	Nasopharyngeal swab	Zymo Quick-RNA Viral	16.2	12,670,186
RAPID13	Influenza A	pdmH1N1	Viral culture	Zymo Quick-RNA Viral	29.8	15,200,408
RAPID14		pdmH1N1	Viral culture	Zymo Quick-RNA Viral	3.5	12,405,816
RAPID15	Mock community	None	Mixed culture	ZymoBIOMICS DNA/RNA Miniprep	44.5	13,541,676
RAPID16		None	Mixed culture	ZymoBIOMICS DNA/RNA Miniprep	86	12,427,300
RAPID17	Kids SARI	Unknown	Nasopharyngeal aspirate	ZymoBIOMICS DNA/RNA Miniprep	6.6	16,321,598
RAPID18		Unknown	Nasopharyngeal aspirate	ZymoBIOMICS DNA/RNA Miniprep	1.5	17,340,092
RAPID19		Unknown	Nasopharyngeal aspirate	ZymoBIOMICS DNA/RNA Miniprep	1	15,464,422
RAPID20		Unknown	Nasopharyngeal aspirate	ZymoBIOMICS DNA/RNA Miniprep	8.3	16,563,150
RAPID21		Unknown	Nasopharyngeal aspirate	ZymoBIOMICS DNA/RNA Miniprep	1.6	24,661,800
RAPID22		Unknown	Nasopharyngeal aspirate	ZymoBIOMICS DNA/RNA Miniprep	8.8	14,490,708
RAPID23		Unknown	Nasopharyngeal aspirate	ZymoBIOMICS DNA/RNA Miniprep	3.3	28,187,178
RAPID24		Unknown	Nasopharyngeal aspirate	ZymoBIOMICS DNA/RNA Miniprep	3.8	19,042,138
RAPID25	RSV	RSV	Nasopharyngeal swab	Roche MagNA Pure 96 Viral NA	84.2	15,287,586
RAPID26		RSV	Nasopharyngeal swab	Roche MagNA Pure 96 Viral NA	94.1	19,473,790
RAPID27		RSV	Nasopharyngeal swab	Roche MagNA Pure 96 Viral NA	97.5	16,302,456
RAPID28		RSV	Nasopharyngeal swab	Roche MagNA Pure 96 Viral NA	80.1	14,524,914
RAPID29		RSV	Nasopharyngeal swab	Roche MagNA Pure 96 Viral NA	67.9	17,923,728
RAPID30		RSV	Nasopharyngeal swab	Roche MagNA Pure 96 Viral NA	49.3	13,466,538
RAPID31		RSV	Nasopharyngeal swab	Roche MagNA Pure 96 Viral NA	56.7	12,192,164
RAPID32		RSV	Nasopharyngeal swab	Roche MagNA Pure 96 Viral NA	61.8	17,199,224
RAPID33	Kids unknown	Unknown	Nasopharyngeal aspirate	ZymoBIOMICS DNA/RNA Miniprep	22.2	14,839,480
RAPID34		Unknown	Nasopharyngeal swab	Zymo Quick-RNA Viral	31.4	14,806,470
RAPID35		Unknown	Nasopharyngeal swab	Zymo Quick-RNA Viral	70.1	12,576,818
RAPID36		Unknown	Nasopharyngeal aspirate	Zymo Quick-RNA Viral	10.9	13,418,996
RAPID37		Unknown	Vomitus	Zymo Quick-RNA Viral	11.8	9,862,060
RAPID38		Unknown	Nasopharyngeal swab	Zymo Quick-RNA Viral	48.1	11,384,408
RAPID39		Unknown	Nasopharyngeal swab	Zymo Quick-RNA Viral	134.5	20,986,506
RAPID40	NTC	NTC	Water	N/A	6.4	26,137,100

**Table 4 viruses-15-01006-t004:** Log-transformed reads per million (_log_RPM) virus distribution across RAPID*prep* samples, with red shading indicating greater read depth as per the key provided at the bottom of the table.

Library.	Virus	Type ^#^	SARS-CoV2	RSV-A	RSV-B	Flu-A pdmH1N1	Flu-C	Rhinovirus	GB Virus C	CMV	HHV7
RAPID01	SARS-CoV-2	NP swab	5.80	0.00	0.00	0.00	0.00	0.00	0.00	0.00	0.00
RAPID02	SARS-CoV-2	NP swab	3.49	0.00	0.00	0.00	0.00	0.00	0.00	0.00	0.00
RAPID03	SARS-CoV-2	NP swab	3.50	0.00	0.00	0.00	0.00	0.00	0.00	0.00	0.00
RAPID04	SARS-CoV-2	NP swab	5.89	0.00	0.00	0.00	0.00	0.00	0.00	0.00	0.00
RAPID05	SARS-CoV-2	NP swab	3.98	0.00	0.00	0.00	0.00	0.00	0.00	0.00	0.00
RAPID06	SARS-CoV-2	NP swab	5.92	0.00	0.00	0.00	0.00	0.00	0.00	0.00	0.00
RAPID07	SARS-CoV-2	NP swab	5.82	0.00	0.00	0.00	0.00	0.00	1.15	0.00	0.00
RAPID08	SARS-CoV-2	NP swab	5.93	0.00	0.00	0.00	0.00	0.00	0.00	0.00	0.00
RAPID09	SARS-CoV-2	NP swab	5.98	0.00	0.00	0.00	0.00	0.00	0.00	0.00	0.00
RAPID10	SARS-CoV-2	NP swab	4.90	0.00	0.00	0.00	0.00	0.00	0.00	0.00	0.00
RAPID11	SARS-CoV-2	NP swab	4.81	0.00	0.00	0.00	0.00	0.00	0.00	0.00	0.00
RAPID12	SARS-CoV-2	NP swab	5.98	0.00	0.00	0.00	0.00	0.00	0.00	0.00	0.00
RAPID13	pdmH1N1	Culture	0.00	0.00	0.00	5.94	0.00	0.00	0.00	0.00	0.00
RAPID14	pdmH1N1	Culture	0.00	0.00	0.00	5.83	0.00	0.00	0.00	0.00	0.00
RAPID15	None	Culture	0.00	0.00	0.00	0.00	0.00	0.00	0.00	0.00	0.00
RAPID16	None	Culture	0.00	0.00	0.00	0.00	0.00	0.00	0.00	0.00	0.00
RAPID17	Unknown	NP aspirate	0.00	0.00	0.00	0.00	0.00	0.00	0.00	0.62	0.00
RAPID18	Unknown	NP aspirate	0.00	0.00	0.00	0.00	0.00	0.00	0.00	0.00	0.00
RAPID19	Unknown	NP aspirate	0.00	0.00	0.00	0.00	0.00	0.00	0.00	0.00	0.00
RAPID20	Unknown	NP aspirate	0.00	0.00	0.00	0.00	0.00	0.00	0.00	0.00	0.00
RAPID21	Unknown	NP aspirate	0.00	0.00	0.00	0.00	0.00	0.00	0.00	0.00	0.00
RAPID22	Unknown	NP aspirate	0.00	0.00	0.00	0.00	5.77	0.00	0.00	0.00	0.00
RAPID23	Unknown	NP aspirate	0.00	0.00	0.00	0.00	0.00	0.00	0.00	0.00	0.00
RAPID24	Unknown	NP aspirate	0.00	0.00	0.00	0.00	0.00	0.00	0.00	0.00	0.00
RAPID25	RSV	NP swab	0.00	0.00	2.64	0.00	0.00	0.00	0.00	0.00	0.00
RAPID26	RSV	NP swab	0.00	3.96	0.00	0.00	0.00	0.00	0.00	0.00	0.00
RAPID27	RSV	NP swab	0.00	0.00	3.25	0.00	0.00	0.00	0.00	0.00	0.00
RAPID28	RSV	NP swab	0.00	2.89	0.00	0.00	0.00	0.00	0.00	0.00	0.00
RAPID29	RSV	NP swab	0.00	0.00	4.09	0.00	0.00	0.00	0.00	0.00	0.00
RAPID30	RSV	NP swab	0.00	5.25	0.00	0.00	0.00	0.00	0.00	0.00	0.00
RAPID31	RSV	NP swab	0.00	4.67	0.00	0.00	0.00	0.00	0.00	0.00	0.00
RAPID32	RSV	NP swab	0.00	0.00	3.14	0.00	0.00	0.00	0.00	0.00	0.00
RAPID33	Unknown	NP aspirate	0.00	0.00	0.00	0.00	0.00	5.61	0.00	0.00	0.00
RAPID34	Unknown	NP swab	0.00	0.00	0.00	0.00	0.00	5.94	0.00	0.00	0.00
RAPID35	Unknown	NP swab	0.00	0.00	0.00	0.00	0.00	6.00	0.00	0.00	0.00
RAPID36	Unknown	NP aspirate	0.00	0.00	0.00	0.00	0.00	0.00	0.00	0.00	2.35
RAPID37	Unknown	Vomitus	0.00	0.00	0.00	0.00	0.00	0.00	0.00	0.00	0.00
RAPID38	Unknown	NP swab	0.00	0.00	0.00	0.00	0.00	4.79	0.00	0.00	0.00
RAPID39	Unknown	NP swab	0.00	0.00	0.00	0.00	0.00	5.98	0.00	0.00	0.00
RAPID40	NTC	Water	0.00	0.00	0.00	0.00	0.00	0.00	0.00	0.00	0.00
Log-RPM	6.00	5.00	4.00	3.00	2.00	1.00	0.00				

^#^ NP = Nasopharyngeal.

## Data Availability

Metagenomic sequence libraries used in this study have been submitted to the NCBI short-read archive (SRA) with accession numbers: SRR22726217 SRR22726256.

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
