# Peer review of "RAPIDprep: A Simple, Fast Protocol for RNA Metagenomic Sequencing of Clinical Samples"

_viruses, 2023, doi:10.3390/v15041006_

Round 1
Reviewer 1 Report
In this paper, the authors optimized procedures for identification of current unknown respiratory pathogens, with faster speed and sophisticated target-based sequencing and bioinformatics workflow. They also performed laboratory improvements against the real-world pathogenetic identification. In general, the paper is valuable to the fields. However, some concerns need to be clarified or solved.
Specific comments:
Line3. The title could be changed to “RAPIDprep: A simple, fast protocol for RNA metagenomic sequencing of respiratory clinical samples” since the major efforts were put on optimization of respiratory-related symptom viral identification.
Line199. Since the latest update of SortMeRNA was in August 2021 and the rRNA removal rate is critical for downstream analysis, please specify the version of rRNA databases (SILVA, RFAM) that authors applied in rRNA filtering procedure.
Line201. Please specify the threshold used in BLAST+ and diamond (i.e., evalue, bitscore, identity, other parameters that are important in contig annotation). Also, please specify the GenBank release version.
Line300. Tagmenention should be replaced with “tagmentation”.
Line395-400. Good point.
Line487, and supplementary figure 1. Both table and figure indicated multiple genotypes/viral species were detected using RAPIDprep assay. Whether the RAPIDprep assay is capable of handling with a single sample for multiple RNA viral pathogens with sufficient genome coverage and sequencing depth? Authors may consider adding an extra group concerning multiple genotypes (e.g., XBB1.5 and BA5 mixed samples) to testify the sensitivity of RAPIDprep assay. This is important because 1) during the COVID pandemic, variable genotypes may occur in single clinical samples, and it is important to distinguish emerging predominant genotype with fast speed, and 2) screening potential respiratory symptom related RNA viruses in immune-deficiency, long-hospitalized patients (usually these samples contain more than one potential pathogens) is in urgent need.
Line487. Please add an extra column introducing the sample origin prior to “Library” column.
Line110, 469-478, and supplementary table 2. The authors did an excellent work in optimizing nucleic acid extraction, sequencing and bioinformatics workflow. However, the initial viral load in samples is as important as downstream assay. I noticed that the authors performed qPCR for SARS-CoV-2 and RSV and made an estimation between relative abundance and Ct value. It is indeed a good argument. However, this is reference-based estimation of initial viral load. What if the pathogens are totally unknown?
The authors mentioned a cohort from children with acute respiratory illness with two symptoms (mild and severe), and the symptoms may act as an indicator reflecting the potential viral load in sampling. It would be much appreciated that they could show the state of hospitalized children and the sampling condition, e.g., initial sampling time after the symptoms were diagnosed, the preserved time, in which manner the sample was treated. These may contribute to further optimization of RAPIDprep in facing with unknown pathogens.
Line567, 571, and 605. Please specify the references (journal, DOI, and website)
Reviewer 2 Report
The manuscript by Tulloch et al outlines the design of a metagenomics pipeline for RNA viruses from respiratory samples. The Pipeline effectively uses the backbone of an established shotgun sequencing pipeline with some modifications (including host and rRNA depletion strategies) to identify respiratory viruses within 40 clinical samples. The emphasis on the reagents selected at each step being focused on the provision of a quick turnaround, thereby making it suitable for infectious disease surveillance. The study is scientifically sound and provides some interesting data on the impact of rRNA depletion and other reagents in the preparation of metagenomics libraries. I believe that the study is suitable for publication but have several minor comments that should be addressed by the authors.
Minor
1. 1. Could the authors comment on the potential for removal of viral sequences using host depletion as part of the bioinformatics analysis (L198-199). Any areas conserved between the host and viral genome will result in those reads being removed. Whilst this does not appear to impact upon SARs-CoV-2, Influenza and RSV the protocol is of a metagenomic nature, thereby this possibility cannot be excluded for other viral pathogens. This should at least be reflected on, in the text.
2. 2. A reference should be provided to support the statement by the authors that ribosomal depletion is important in the identification and genome recovery of Coronaviruses, Paramyxoviruses and Influenza viruses (L230-232). If a reference cannot be inserted into the text to support this, the sentence should be omitted.
3. 3. The sentence on L255-257 quotes figure 1A. This follows on from the previous sentence that also quotes figure 1A. One of these should be removed due to their close proximity within the manuscript.
4. 4. Optimization of a metagenomics protocol is difficult as you do not necessarily know what pathogen is in the sample a proiri. Whilst the RAPIDprep protocol is aimed generally at respiratory RNA included those in the study the optimization here is only performed on samples spiked with a single dilution of Influenza, SARS-CoV-2 and RSV. This should be emphasized in the text that these levels of amplification will be specific for these viruses and could potentially differ for other virus types, depending upon viral load
5. 5. Table 3 outlines the details of how the clinical samples used in this study were treated but it is difficult to contrast and compare between the distinct groups as they have been processed differently, particularly with regard to nucleic acid extraction and sample types. Could the authors comment on how these different extraction kits could potentially account for some of the differences between samples. Details including different extraction methods should be added to Figure 2 to help readers understand any comparisons. Are the Kids SARI samples showing different compositions because of the extraction or is it a true impact of the multiple freeze thaw of the samples. Surely, these effects would be evident in other sample types also? The RSV samples look similar and different to the other samples – is it down to extraction or another reason. Whist the authors may be right in their conclusions in the manuscript, this variability makes it difficult to fully grasp what this figure conveys.
Reviewer 3 Report
The manuscript by Tulloch et al. describes an RNA seq method that includes DNAse treatment, rRNA depletion, cDNA synthesis, ss synthesis, tagmentation based library prep, and Illumina sequencing. Some steps of the protocol have been tested on 3 respiratory samples and the proof-of-concept was obtained by applying the method on other samples (clinical, cell culture supernatant, standards). The method works as expected. While I think that the method has been properly designed and nicely presented, it looks like the authors are overselling a basic protocol that is very similar to what is widely used for metagenomic studies all over the globe. The method is not innovative and, out of the 3 steps they tried to tweak, 2 didn’t work and one (adjusting the number of PCR cycles during library prep) is something that is already normally standardized in many laboratories depending on DNA concentration. Additionally, the authors do not talk about the disadvantages of the method (the biggest one being not being able to detect DNA viruses – unless it’s mRNA - which is kind of important if they are trying to sell it as a catch-all method for diagnostics), are not entirely fair with the literature cited, and I have some doubts about what they call “optimization”. I think the paper is of good enough quality to be published after some of the concept presented and statements made are tuned down to provide an honest presentation of the methods. Specific comments:
1. I have some doubts about the first part of the study where they say they optimized the method. To call this “optimization” they should have tested virus-positive samples and assessed virus detectability under various conditions, besides monitoring the background. Especially, RNA depletion methods are known to reduce the concentration of viral RNA too… without assessing virus detectability in the different protocols they can only control for the reverse transcription of the background but can’t affirm that they found the best conditions for the protocol. Therefore, this cannot be called optimization. Unless the authors have virus data to show, wording needs to be adjusted and this has to be stated clearly as study limitation.
2. The only modification that seemed to improve the basic protocol was the number of PCR cycles after adaptor ligation. Since the fine-tuning of the number of PCR cycles to perform to obtain a high concentration of good quality DNA is largely dependent on the DNA concentration after adaptor ligation, I am wondering why this wasn’t measured before the PCR and results evaluated also considering this parameter. If the authors want to state that this is concentration-independent and works for all samples, concentrations should be provided to demonstrate it.
3. Study limitations should be clearly stated in the discussion and in the abstract, especially the fact that the test is not able to detect DNA viruses.
4. In the conclusions you mention that this method was successfully used before to identify viruses. However, in the cited literature, viruses were identified by PCR and the complete genomes were obtained with a different method. This is not a truthful description of the method’s application.
5. Do the 24 hours include sequencing and data analyses? Maybe you should break down the timing from DNA isolation to results.
Minor:
-Line 30. “using a single protocol” is a weird expression as 1 single protocol can include any number of steps.
-Lines 39-46. More references are needed in this paragraph.
-Lines 68-78. More references are needed in this paragraph.
-Line 82. This sentence is a bit odd, shouldn’t it be “RNA pathogens in samples in less than 24 hours”?
-Tables 2 and 4. I am assuming that the last line is the color-legend. You should probably state this better because it is not entirely clear.
-You should probably mention how the read duplication rate is calculated.
-Lines 377-9. Shouldn’t it be the other way around (mapped reads divided by the total number of reads)?
-Line 388. Typo: nucleic acids.
-Lines 541-2. You forgot to fill in this part.
Reviewer 4 Report
The manuscript by Rachel et al described an RNA metagenomic sequencing protocol for detection of RNA viruses in clinical samples. The study generated data for optimizing three key steps of the protocol. The study was well designed, and the manuscript was well written. The information presented will be helpful for the clinical laboratories that are interested in applying the technology in patient care.
Comments
1. The title is misleading. The protocol took the common NGS approach. There was no major innovation in simplifying the process, and the assay time was not significantly reduced. The protocol failed to deplete human genomic DNA efficiently which is the major problem with sequencing clinical samples. Although the protocol lacks novelty, the approach presented in the study to optimize the assay is valuable for the clinical labs that are in the process of developing in-house assays.
2. Pre-tested clinical samples were used for the assay development. No information on the clinical testing methods was provided. The information would be important for evaluating the quality of the RAPIDprep assay results.
3. Figure 3B showed detection of RSV with the RAPIDprep assay in the samples with a relative high RSV viral load (Ct <25). Will be the assay be successful for the samples with low viral load?
4. As an assay for clinical testing, it’s important to develop standardized criteria for acceptance of quality results. For RAPIDprep assay, the authors stated that target yields of >1 M non-human, non-rRNA would be ideal for generating reliable result. However, RAPID 30 only had 5,898 read non-human non-rRNA reads, but the assay successful detected RSV in the sample. What should be used as the assay quality check?
5. Line 439-442, the RAPIDprep produced positive results in samples that previously tested negative. One challenge of NGS for pathogen detection for clinical use is to determine the clinical relevance of the result. Because of the high sensitivity of the technology, it’s common for NGS based assay to detect benign colonizer with no clinical significance. As an assay for clinical use, it’s important to establish criteria for RAPIDprep to call clinically relevant results. This information was not provided in the manuscript.
6. After gDNA removal and rRNA depletion, how can the RAPIDprep assay be used for the detection of bacterial in clinical samples as the data presented in figure 4? What’s the purpose of 3.4 study?
Round 2
Reviewer 1 Report
The authors have addressed my concerns, and I have no more comments on it.
Reviewer 3 Report
I am satisfied with the revisions.